METHODS AND PROTOCOLS

# Powerful and Real-Time Quantification of Antifungal Efficacy against Triazole-Resistant and -Susceptible *Aspergillus fumigatus* Infections in *Galleria mellonella* by Longitudinal Bioluminescence Imaging

Eliane Vanhoffelen,[a] Lauren Michiels,[a] Matthias Brock,[b] Katrien Lagrou,[c,d] Agustin Reséndiz-Sharpe,[a] Greetje Vande Velde[a]

[a]Department of Imaging and Pathology, Biomedical MRI Unit/MoSAIC, KU Leuven, Leuven, Belgium
[b]School of Life Sciences, Fungal Biology Group, University of Nottingham, Nottingham, United Kingdom
[c]Department of Microbiology, Immunology and Transplantation, Laboratory of Clinical Bacteriology and Mycology, KU Leuven, Leuven, Belgium
[d]Department of Laboratory Medicine, National Reference Center for Mycosis, University Hospitals Leuven, Leuven, Belgium

**ABSTRACT** *Aspergillus fumigatus* is an environmental mold that causes life-threatening respiratory infections in immunocompromised patients. The plateaued effectiveness of antifungal therapy and the increasing prevalence of triazole-resistant isolates have led to an urgent need to optimize and expand the current treatment options. For the transition of *in vitro* research to *in vivo* models in the time- and resource-consuming preclinical drug development pipeline, *Galleria mellonella* larvae have been introduced as a valuable *in vivo* screening intermediate. Despite the high potential of this model, the current readouts of fungal infections in *G. mellonella* are insensitive, irreproducible, or invasive. To optimize this model, we aimed for the longitudinal quantification of the *A. fumigatus* burden in *G. mellonella* using noninvasive bioluminescence imaging (BLI). Larvae were infected with *A. fumigatus* strains expressing a red-shifted firefly luciferase, and the substrate dosage was optimized for the longitudinal visualization of the fungal burden without affecting larval health. The resulting photon flux was successfully validated for fungal quantification against colony forming units (CFU) analyses, which revealed an increased dynamic range from BLI detection. Comparison of BLI to survival rates and health index scores additionally revealed improved sensitivity for the early discrimination of differences in fungal burdens as early as 1 day after infection. This was confirmed by the improved detection of treatment efficacy against triazole-susceptible and -resistant strains. In conclusion, we established a refined *G. mellonella* aspergillosis model that enables the noninvasive real-time quantification of *A. fumigatus* by BLI. This model provides a quick and reproducible *in vivo* system for the evaluation of treatment options and is in line with 3Rs recommendations.

**IMPORTANCE** Triazole-resistant *Aspergillus fumigatus* strains are rapidly emerging, and resistant infections are difficult to treat, causing mortality rates of up to 88%. The recent WHO priority list underscores *A. fumigatus* as one of the most critical fungal pathogens for which innovative antifungal treatment should be (urgently) prioritized. Here, we deliver a *Galleria mellonella* model for triazole-susceptible and -resistant *A. fumigatus* infections combined with a statistically powerful quantitative, longitudinal readout of the *A. fumigatus* burden for optimized preclinical antifungal screening. *G. mellonella* larvae are a convenient invertebrate model for *in vivo* antifungal screenings, but so far, the model has been limited by variable and insensitive observational readouts. We show that bioluminescence imaging-based fungal burden quantification outperforms these readouts in reliability, sensitivity, and time to the detection of treatment effects in both triazole-susceptible and -resistant infections and can thus lead to better translatability from *in vitro* antifungal screening results to *in vivo* confirmation in mouse and human studies.

Address correspondence to Greetje Vande Velde, greetje.vandevelde@kuleuven.be.

The authors declare a conflict of interest. Katrien Lagrou received consultancy fees from MRM Health and MSD, speaker fees from Pfizer and Gilead and a service fee from Thermo Fisher Scientific and TECOmedical. The remaining authors report no conflict of interest.

**KEYWORDS** *Galleria mellonella*, *Aspergillus fumigatus*, antifungal screening, bioluminescence imaging, triazole resistance, antifungal therapy, BLI, aspergillosis, fungal infection

Fungal infections are an emerging public health concern leading to significant morbidity and mortality worldwide, especially among the growing immunocompromised population (1). Currently, systemically available antifungal compounds are limited to four classes of antifungals, i.e., triazoles, echinocandins, pyrimidines, and polyenes. While these treatments are effective, they are often associated with adverse effects and frequent drug-drug interactions, complicating their use (1). Moreover, antifungal resistance against existing compounds is rapidly emerging, reinforcing the need for novel antifungal treatments (1).

*Aspergillus fumigatus* (AF) was listed by the WHO as one of the four most critical fungal pathogens for which innovative antifungal treatments are urgently required (1). This ubiquitous environmental mold can cause a severe form of disease known as invasive pulmonary aspergillosis (IPA), affecting more than 300,000 people annually (2). Triazoles are the recommended first-line treatment for IPA, but triazole-resistant AF isolates are emerging, with prevalences ranging from 5.4% in Belgium to 17.8% in the United Kingdom and up to 80% in China (3–5). The most commonly isolated triazole-resistant AF strains have a $TR_{34}$/L98H or a $TR_{46}$/Y121F/T289A mutation in the *cyp51A* gene, encoding an enzyme involved in the production of ergosterol (6). Mortality rates associated with triazole-resistant AF infections are alarmingly high, ranging from 47 to 88%, with an excess overall mortality rate of 21% compared to triazole-susceptible cases (1, 7, 8). This highlights the urgent need for novel antifungals against triazole-resistant AF isolates.

The typical pipeline for preclinical antifungal drug discovery starts with *in vitro* screening for the antifungal activity of novel compounds or synergism between existing compounds followed by *in vivo* testing in mouse models. Although mouse models are still indispensable for drug discovery (9), they are far from ideal for performing high-throughput screenings for potential antifungal compounds. Ethical restrictions, the large number of animals necessary to obtain sufficient statistical power, high costs, and labor-intensiveness make them a time- and resource-consuming research step (10, 11). Moreover, many compounds showing antifungal activity *in vitro* fail once they are tested in mice because of the additional layer of complexity of *in vivo* processes (9). Mouse models of invasive aspergillosis are especially complex since they require immunosuppression and are often associated with severe animal welfare impairment by using mortality or euthanasia in a moribund state as the endpoint. Therefore, it is of the utmost importance to limit the number of mice involved in antifungal screening (9, 12).

Over the last decades, larvae of *Galleria mellonella* (the greater wax moth) have been increasingly used as a preclinical infection model, as illustrated by the exponential increase in published studies using these larvae over the last 10 years. The larvae of *Galleria mellonella* are susceptible to fungal and bacterial infections, enabling studies of the virulence of microbial strains, host-pathogen interactions, and antimicrobial treatment efficacy (13–15). Moreover, virulence and survival outcomes frequently correlate well between *G. mellonella* and mouse models (16–20). The main advantages of this invertebrate model over traditional murine models are its low cost, ease of use, ease of ethical and biosafety regulations, and convenience in testing many different experimental conditions in a time-efficient way. In contrast to mice, *G. mellonella* larvae are insensitive to pain because they lack nociceptors, and therefore, no ethical restrictions exist (21, 22).

As living hosts, *G. mellonella* larvae offer significant advantages over *in vitro* systems, mainly because they have an innate immune system including both a cellular and a humoral compound, similar to vertebrates (21). Compared to other popular invertebrate models such as *Caenorhabditis elegans* or *Drosophila melanogaster*, *G. mellonella* larvae are unique in that they withstand a temperature of 37℃, allowing the activation of temperature-dependent virulence factors of human pathogens (21). Altogether, this makes *G. mellonella* a relevant host for the efficient testing and selection of antifungal compounds in an *in vivo* model before eventually moving to murine models for even more complex investigations of the

selected compounds (23). *G. mellonella* thereby bridges the translational gap between *in vitro* and mouse studies and can limit the untimely use of mouse models, thus complying with the replacement, reduction and refinement (3Rs) and contributing to a more efficient preclinical drug development pipeline.

Indeed, several studies have confirmed the relevance of *G. mellonella* larvae in evaluating the efficacy of antifungal drugs against AF infection (24). They have been used to determine the pharmacokinetic profiles of clinically available triazoles and amphotericin B (AMB) in the hemolymph and to assess the *in vivo* efficacy of these compounds against several triazole-susceptible and -resistant AF isolates (11, 25). A new compound, hemofungin, was successfully tested for antifungal activity *in vitro* and in larvae (26). Also, synergism between compounds has been tested in *G. mellonella*, e.g., itraconazole and EGTA (a calcium chelator) or azoles combined with pyrvinium pamoate (an antiparasitic drug) or AZD8055 (an antitumor agent) (27–29). Furthermore, in the context of drug repurposing, sertraline (an antidepressant) improved survival in AF-infected larvae. This was further validated in mice, showing a reduced pulmonary fungal burden (30). These studies show that *G. mellonella* larvae have proven useful in the context of antifungal drug discovery and that the results are in line with those from *in vitro* and mouse models.

Despite the many advantages of the *G. mellonella* model, large variability exists within and between experiments, which is an important factor keeping this model from being widely used. On the one hand, this variation is caused largely by the varying conditions and qualities of the larvae, fueled by the widespread lack of available research-grade *G. mellonella* larvae (21, 31), especially in the post-Brexit regulatory landscape in Europe, where the only company providing research-grade larvae could no longer dispatch them to customers' research laboratories. Therefore, most researchers using *G. mellonella* larvae are purchasing them from commercial reptile feed or fishing bait suppliers, where breeding conditions are not regulated and larvae have often traveled long distances from the wholesaler to local stores under uncontrolled shipping conditions. These larvae may already carry infections, and it has been shown that stress, temperature, and delays during transportation significantly affect larval health (22). This source of variability could be minimized by providing research-grade larvae with standardized, quality-controlled rearing and dispatching protocols to the research community. On the other hand, larval health and survival are still the most widely used readouts in *G. mellonella* studies. Consequently, the use of health scores as a singular readout cannot differentiate between the health effects caused by experimental infection and those caused by other factors such as coinfections. Moreover, health scoring is also prone to interobserver variability, ultimately hampering a reliable comparison of results from the literature. Other commonly used readouts of the fungal burden in *G. mellonella* are colony forming units (CFU) and histopathology, providing objective data independent of larval health, but these are labor-intensive and static endpoint measurements that are unable to unravel the dynamic aspects of infection. Additional dynamic readouts that bypass health parameters are thus necessary to unlock the full potential of this promising model for antifungal screening against resistant *A. fumigatus* strains. To that end, we propose bioluminescence imaging (BLI), which is a noninvasive, longitudinal method to directly quantify the fungal burden *in vivo* that is already used in mouse models of IPA (32). The translation of this method to *G. mellonella* could overcome many of the above-mentioned limitations and can provide additional quantitative information on fungal burdens in real time.

In this study, we aimed to establish the first *G. mellonella* model of triazole-resistant and -susceptible aspergillosis with the direct and noninvasive quantification of the fungal burden over time based on BLI. By using BLI, we aimed to combine the objectivity of CFU determination and the longitudinal noninvasive aspect of health scoring, creating a quick and reproducible screening tool for antifungal drug testing against triazole-susceptible and -resistant AF strains that have the same genetic background (32) to avoid the impact of strain variation on the results. We compared our BLI readout to survival, health score, and CFU readouts and benchmarked the model for antifungal therapy screening using first-line antifungals against triazole-susceptible and -resistant AF strains.

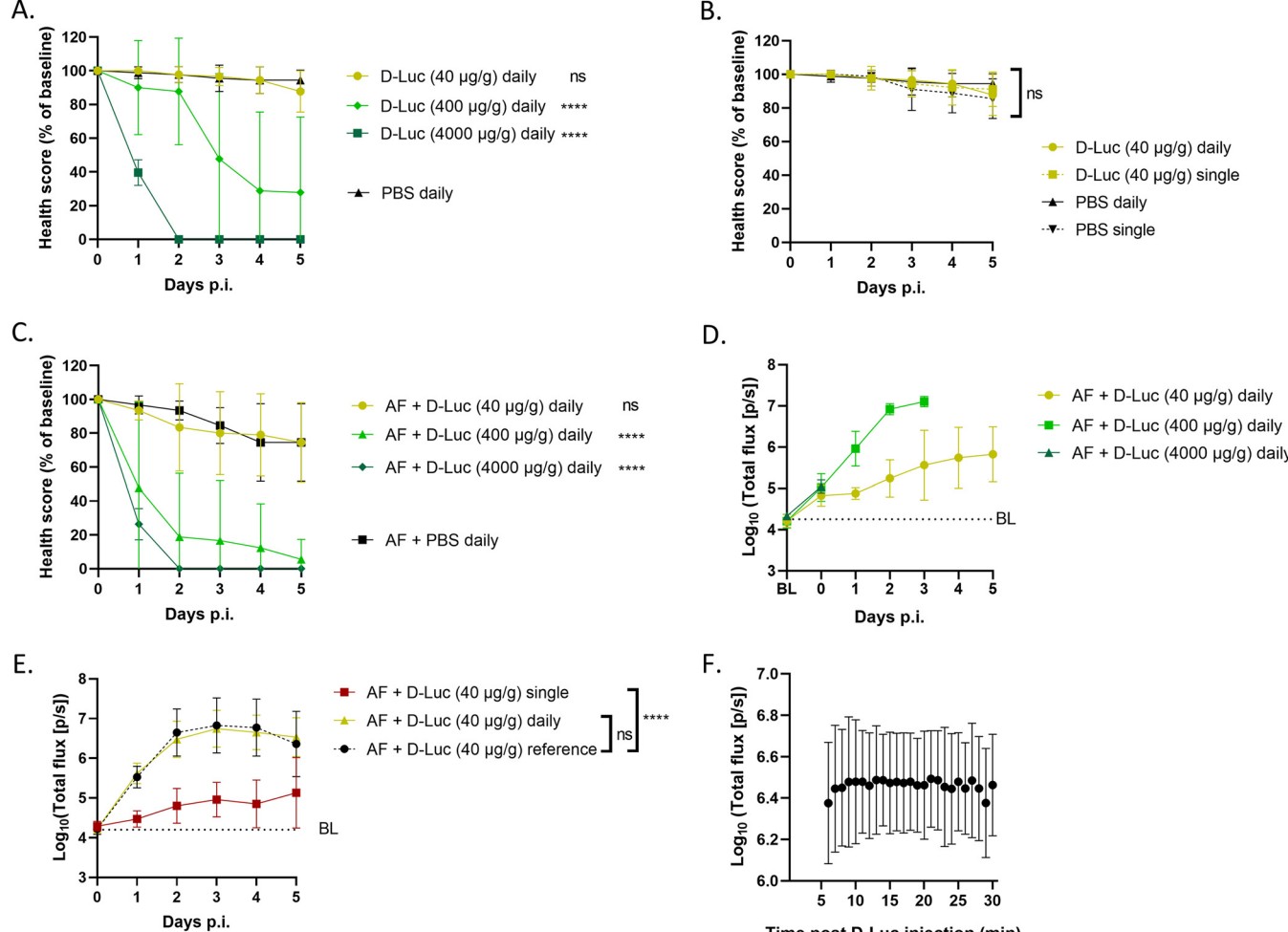

**FIG 1** Optimization of *in vivo* BLI in *G. mellonella*: tolerability, dose, administration, and kinetics of the substrate. (A and B) Tolerability of different doses of daily D-luciferin (D-Luc) injections (A) and single and daily D-luciferin injections (40 μg/g) (B), compared to the corresponding PBS group, in healthy larvae measured by health scores ($n = 10$). (C and D) Tolerability of different doses of daily D-luciferin injections compared to PBS in larvae infected with $10^5$ conidia of AF TR$_{34}$/L98H measured by health scores (C) and the resulting longitudinal *in vivo* BLI signals (D) ($n = 10$). Longitudinal statistical analysis could not be performed because of the different endpoints. (E) *In vivo* BLI signals in larvae infected with $10^5$ conidia of AF TR$_{34}$/L98H receiving either daily D-luciferin injections or a single injection after infection on day 0, compared to equally AF-infected larvae used as a daily cross-sectional reference group receiving only D-luciferin on the day of interest ($n = 20$). (F) BLI signal stability over time in larvae infected with $10^5$ conidia of AF TR$_{34}$/L98H after D-luciferin injection on day 3 postinfection (p.i.) ($n = 10$). BL (baseline) represents the background BLI signal (dotted lines). Data are means ± standard deviations (SD). ****, $P < 0.0001$; ns, nonsignificant.

## RESULTS

**Daily substrate injections to visualize the fungal burden in *G. mellonella* without affecting larval health.** To generate a bioluminescence signal representative of the fungal burden present in individual *G. mellonella* larvae, the luciferase produced within *A. fumigatus* (AF) needs to be in direct contact with the substrate D-luciferin. The most efficient way to do this is by injecting D-luciferin into the hemocoel. A few aspects need to be balanced for this: the tolerability of the larvae to the substrate on the one hand and the optimal substrate dose, frequency of administration, and kinetics to generate a dynamic photon flux range on the other hand. First, we looked for the optimal substrate dose that would be well tolerated by the larvae without inducing toxicity but would also generate a sufficiently high photon flux for the sensitive quantification of the fungal burden. Starting from the optimal D-luciferin dose for bioluminescence imaging (BLI) of AF in mice (500 mg/kg of body weight intraperitoneally [32]), three doses of D-luciferin (40, 400, and 4,000 μg/g) were tested in uninfected *G. mellonella* larvae upon a single or daily injection. The lowest dose (40 μg/g) was the only one that did not cause declines in the health score and survival compared to phosphate-buffered saline (PBS) (Fig. 1A; see also Fig. S1A in the supplemental material). Moreover, the tolerability of daily injections compared to single injections of the 40-μg/g D-luciferin dose

was high as it did not affect larval health more than single injections (Fig. 1B and Fig. S1B). We tested the same three luciferin doses and their tolerability upon daily injection in AF-infected larvae (TR$_{34}$/L98H, $10^5$ conidia/larva), and again, daily 40-$\mu$g/g D-luciferin administration was tolerated the best (Fig. 1C and Fig. S1C). Additionally, daily injections of this substrate dose generated a sufficiently dynamic photon flux above the background to detect *in vivo* fungal growth in infected larvae over multiple days without reaching the detection limit, as shown in the corresponding BLI readout (Fig. 1D).

To investigate which frequency of substrate injection is necessary to generate a photon flux representative of the fungal burden present over time, we compared daily injections to a single injection of 40 $\mu$g/g D-luciferin immediately after infection (day 0). As a cross-sectional control and to exclude any effects of possible D-luciferin accumulation on the photon flux after repeated substrate injections, we added daily reference groups receiving only a single D-luciferin injection on the respective days of BLI scanning. We found that the photon flux generated by daily substrate injection corresponded well with the daily reference groups, indicating that daily injection of D-luciferin is necessary and sufficient to reliably visualize the AF burden present (Fig. 1E). Remarkably, a single injection of D-luciferin on day 0 resulted in photon flux above the baseline until day 5 postinfection (p.i.), suggesting that D-luciferin is not fully metabolized or excreted in *G. mellonella*. However, this potential D-luciferin accumulation over time did not affect the longitudinal photon flux in the groups receiving the substrate daily at 40 $\mu$g/g given its good agreement with the cross-sectional reference curve (Fig. 1E). Finally, we looked at the kinetics of photon flux after the injection of the substrate. The BLI signal in *G. mellonella* stayed at a constant level during the first 30 min after the D-luciferin injection. Therefore, we decided on a time of 10 min between D-luciferin injection and image acquisition for all further acquisitions (Fig. 1F and Fig. S2). In summary, the daily administration of 40 $\mu$g/g D-luciferin via intrahemocoel injection visualized the AF burden in *G. mellonella* well without affecting larval health.

**The *in vivo* BLI signal is a quantitative measure of the fungal burden over time in *G. mellonella*.** To validate the *in vivo* BLI signal in *G. mellonella* for its suitability for fungal quantification, we compared the *in vivo* BLI signal with the postmortem (*ex vivo*) BLI signal and CFU of larval homogenates, whereby the latter readout is currently seen as the gold standard for the quantification of the fungal burden. First, larvae infected with a range of $10^3$ to $10^8$ conidia of the luciferase-expressing AF TR$_{34}$/L98H or wild-type (WT) strains were scanned by BLI at 2 h postinfection and sacrificed for CFU determination immediately after *in vivo* BLI. At the very onset of infection, a comparison of the *in vivo* BLI and CFU measurements revealed that at the lower end of the infectious doses ($10^3$ to $10^5$ conidia), CFU determination outperformed BLI because its detection limit went below the baseline signal of BLI ($\sim$4.5 $\times$ $10^4$ photons/s [p/s] in these data), down to 1.5 CFU/g and possibly even lower (Fig. 2A and B). In contrast, in moderate- to high-dose-infected larvae ($10^5$ to $10^8$ conidia), CFU counts plateaued, while the *in vivo* BLI signal showed a proportional increase (Fig. 2A and B). In this higher range of conidia above the BLI baseline, however, a good correlation was observed between *in vivo* BLI and CFU counts for TR$_{34}$/L98H (Fig. 2C) and the WT (Fig. 2D), confirming the quantitative character of *in vivo* BLI in *G. mellonella* already at 2 h postinfection.

Next, we wanted to confirm this quantitative correlation between both methods longitudinally over the course of infection, which could be affected by increasing melanization and/or hypha formation. We monitored live larvae infected with $10^5$ conidia of AF TR$_{34}$/L98H from day 0 until day 5 postinfection, and cross-sectional comparisons between *in vivo* BLI and CFU counts of the homogenates were performed daily. *In vivo* BLI showed an increase in the fungal load until day 2 p.i., with a stagnation of the signal afterward because of survival bias, while the corresponding CFU counts showed no increase over time at all (Fig. 2E). When all data were pooled over time to assess the overall agreement of both readouts, a clear proportional bias toward higher counts appeared, indicating a larger dynamic range of BLI (Fig. 2F). This larger dynamic range of *in vivo* BLI than of CFU counts is also reflected in the flat correlation between both methods and could be due to a reduced correlation of fungal filaments with CFU counts (Fig. 2H). When comparing *in vivo* BLI to *ex vivo* BLI of the larval

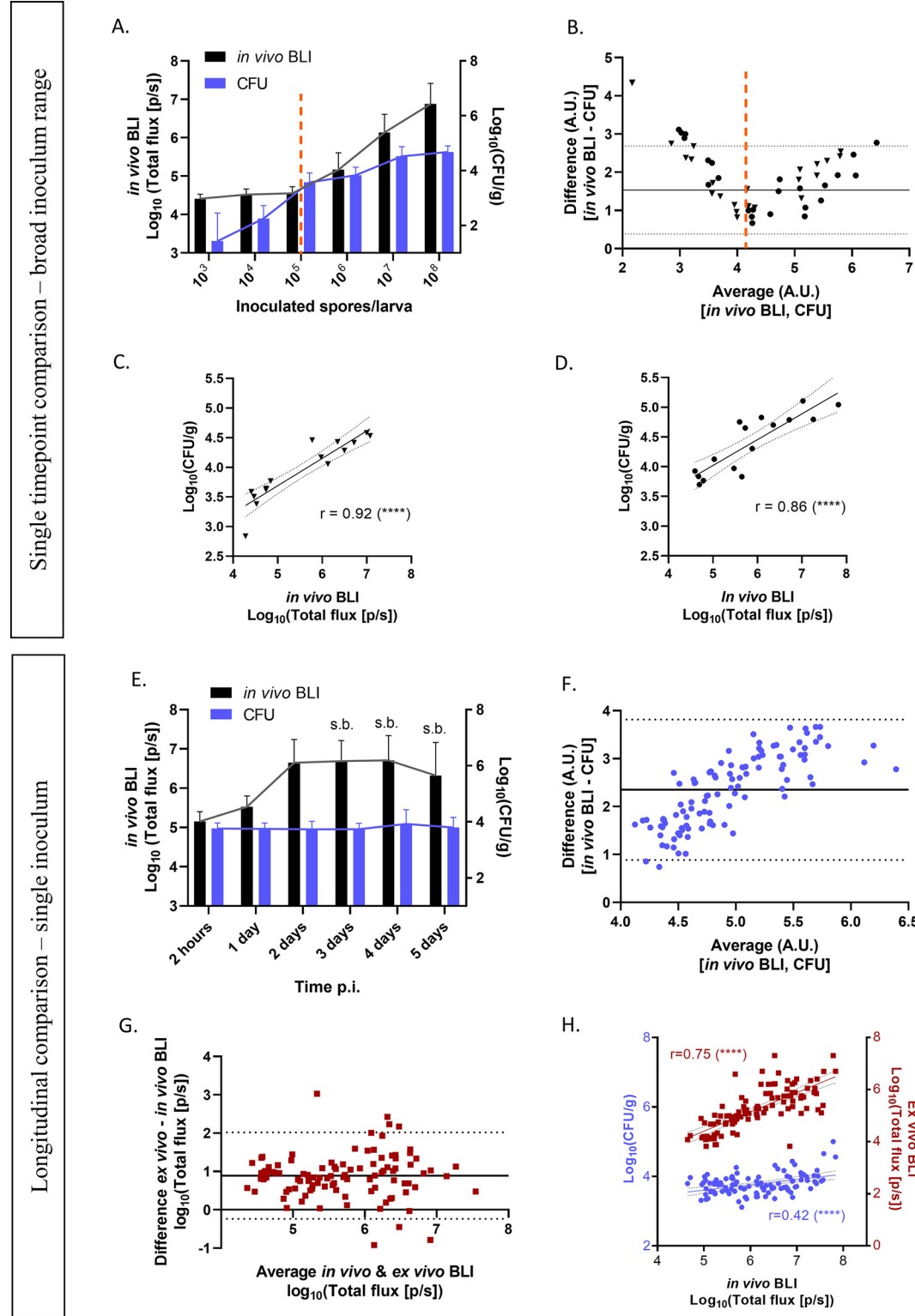

**FIG 2** Methodological agreement and correlation among *in vivo* BLI, *ex vivo* BLI, and CFU in *G. mellonella*. (A and B) Fungal loads in larvae infected with a range of $10^3$ to $10^8$ conidia of AF TR$_{34}$/L98H or the WT (pooled) as measured by *in vivo* BLI or CFU at 2 h p.i. (A) and corresponding Bland-Altman comparisons between *in vivo* BLI signals and CFU for fungal quantification on the day of infection (B). Each data point represents an individual larva infected with $10^3$, $10^4$, $10^5$, $10^6$, $10^7$, or $10^8$ conidia of AF TR$_{34}$/L98H (triangles) or the WT (dots) ($n = 48$); scanned by BLI at 2 h p.i.; and sacrificed for CFU determination immediately thereafter. Both differences and averages are expressed in arbitrary units (A.U.). Proportional linear biases below and above the average of ~$10^4$ arbitrary units (dotted line) indicate higher counts of BLI in the low (<$10^4$ arbitrary units) and high (>$10^4$ arbitrary units) values compared to the CFU, corresponding to panel A. The higher

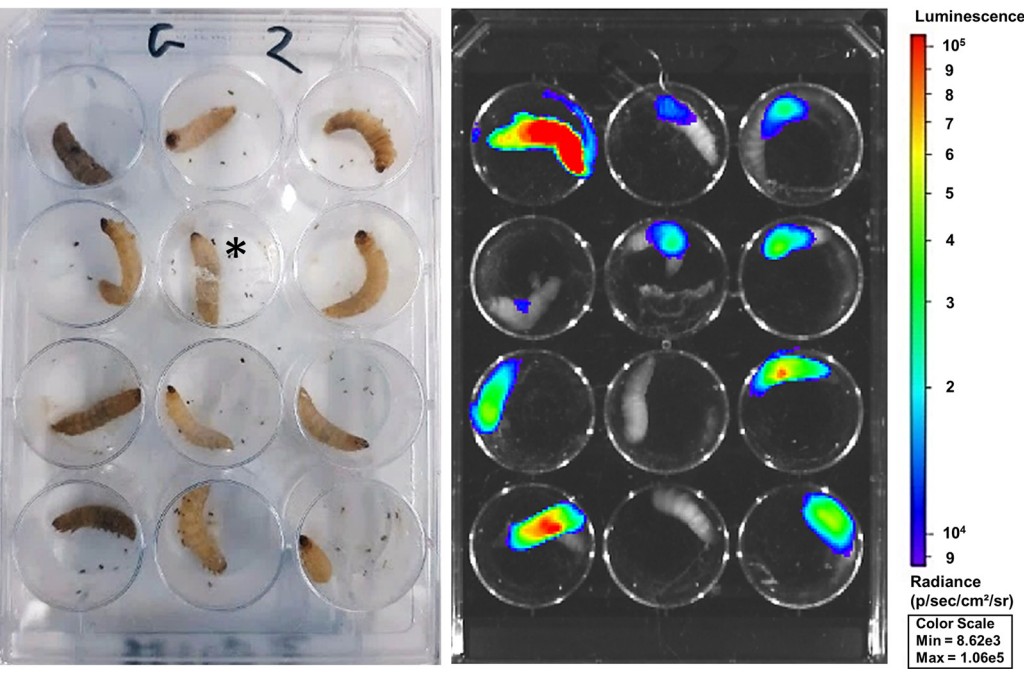

**FIG 3** BLI and health status correlation in *G. mellonella* larvae. Varying health statuses (left) and the corresponding representative BLI overlay (right) are shown. The image depicts larvae ($n = 12$) infected with $10^5$ spores of AF TR$_{34}$/L98H on day 3 postinfection. The bioluminescence signal is visualized within the limits of the scale shown. *, this larva is partially covered by silk, indicating cocoon formation.

homogenates, their differences were equally distributed around the average difference, confirming that the increasing bias between *in vivo* BLI and CFU is not caused by differences between *in vivo* and *ex vivo* samples but has a methodological origin (Fig. 2G). This explains why the correlation coefficient between *in vivo* BLI and CFU is lower than with *ex vivo* BLI (Fig. 2H). In this data set, the dynamic ranges of both *in vivo* BLI and *ex vivo* BLI are 3.2 log units {4.6 to 7.8 $\log_{10}$[total flux (photons/s)]}, compared to 1.9 log units for the corresponding CFU [3.1 to 5.0 $\log_{10}$(CFU/g)] (Fig. 2F). Altogether, these results validate the use of longitudinal *in vivo* BLI as a quantitative measure of the fungal burden in *G. mellonella* over time when photon flux is higher than the background. While CFU determination has a lower detection limit, we show that BLI has a larger dynamic range.

**BLI sensitively discriminates fungal loads *in vivo* in *G. mellonella*.** Next, we compared our longitudinal *in vivo* BLI methodology to health score and survival readouts, currently the only longitudinal readouts of AF infections in *G. mellonella*, looking for sensitive discrimination between different fungal burdens over time. For this, larvae were infected with either $10^3$, $10^4$, or $10^5$ conidia of AF TR$_{34}$/L98H and compared for survival, health score, and *in vivo* BLI readouts over a time course of 5 days. Visually, a lower health score of the larvae generally corresponded to a higher BLI signal (Fig. 3). Over 5 days, the survival and health score readouts were significantly decreased only in the group infected with the highest AF load, with no significant differences at individual time points because of the large standard deviations (Fig. 4A and B). *In vivo* BLI, on the other hand, could significantly distinguish all three

**FIG 2** Legend (Continued)

BLI counts below $10^4$ A.U. can be explained by the background signal and, thus, the limit of detection of the BLI signal of $\sim4.5 \times 10^4$ p/s. Therefore, only *in vivo* BLI signals above the background were included to compute correlation coefficients between CFU and BLI counts in panels C and D. (C and D) Pearson correlations between *in vivo* BLI and CFU of AF TR$_{34}$/L98H ($n = 15$) (C)- and WT AF ($n = 16$)-infected larvae at 2 h p.i. (E) Fungal loads in larvae infected with $10^5$ conidia of AF TR$_{34}$/L98H as measured by *in vivo* BLI or CFU over time (days 0 to 5 p.i.). s.b., survival bias. (F and G) Bland-Altman comparisons between CFU and *in vivo* BLI (F) and *ex vivo* BLI (G) for fungal quantification over time (days 0 to 5 p.i.). Each data point represents an individual larva infected with $10^5$ conidia of AF TR$_{34}$/L98H, scanned by BLI, and cross-sectionally sacrificed for CFU determination and *ex vivo* BLI ($n = 20$ per day). In panel F, both differences and averages are expressed in arbitrary units (A.U.). (H) Pearson correlations between *in vivo* BLI and the respective CFU (blue dots) or *ex vivo* BLI (red squares) in AF TR$_{34}$/L98H-infected larvae from days 0 to 5 p.i. r, Pearson correlation coefficient. All data include only live larvae. ****, $P < 0.0001$.

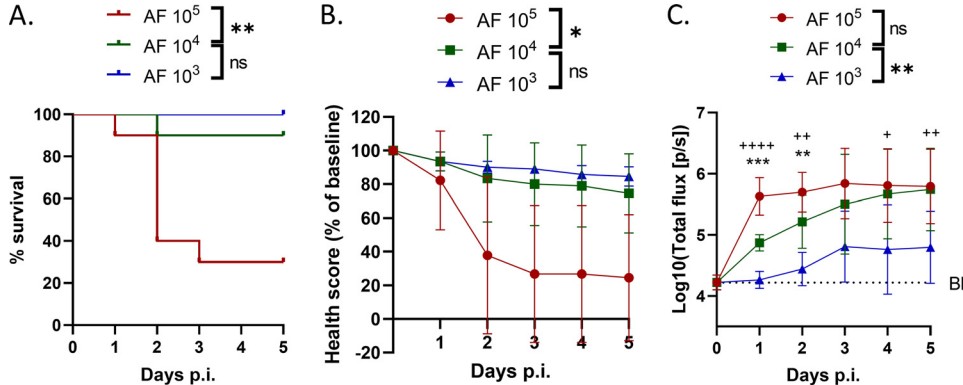

**FIG 4** Comparison among *in vivo* BLI, survival, and health score readouts for discriminating among *Aspergillus fumigatus* (AF) doses over time. Shown are survival (A), health score (B), and BLI signal (C) readouts in larvae infected with $10^3$, $10^4$, or $10^5$ spores of AF TR$_{34}$/L98H over 5 days after infection. Data are means ($\pm$SD) ($n$ = 10). Statistics on the graph refer to differences on individual days with between $10^3$ and $10^4$ conidia per larva ("+") and between $10^4$ and $10^5$ conidia per larva ("*"). Statistics in the keys refer to longitudinal differences over 5 days. */+, $P < 0.05$; **/++, $P < 0.01$; ***/+++, $P < 0.001$; ****/++++, $P < 0.0001$; ns, nonsignificant.

doses of infection as soon as 24 h after infection. Moreover, the photon flux increased over time in an inoculum-dependent manner (Fig. 4C). We conclude that *in vivo* BLI is the most sensitive longitudinal readout as it allows the discrimination of different fungal burdens at the earliest time point.

**BLI enables *in vivo* antifungal efficacy screening under triazole-susceptible and -resistant conditions.** Validation of the BLI readout for antifungal screening against triazole-susceptible and -resistant AF strains was performed using voriconazole (VCZ) and amphotericin B (AMB) treatments, which was again compared to survival and health score readouts. VCZ and AMB treatments as well as their vehicles did not show toxicity effects on sham-infected larvae and were well tolerated (no health score deterioration) (Fig. S3). Larvae infected with $10^5$ conidia of WT AF that received AMB or VCZ had a significantly higher health score (Fig. 5B) than the sham-treated but infected control group, although survival readouts did not show any differences (Fig. 5A). The observed lack of clinical deterioration in the treated groups was in agreement with the lower photon flux from BLI, indicating lower fungal loads (Fig. 5C). BLI was able to detect treatment effects earlier than the health score and was the only readout that was able to differentiate between the AMB- and VCZ-treated groups (Fig. 5C). On day 5 p.i., *in vivo* BLI correlated well with the CFU in larval homogenates (Fig. 5D). As expected, in larvae infected with $10^5$ conidia of triazole-resistant AF TR$_{34}$, a beneficial treatment effect was seen only in AMB- and not VCZ-treated larvae, which was consistently detected by the health score and *in vivo* BLI (Fig. 5F and G) but not by survival analysis (Fig. 5E). Only BLI was able to detect the treatment effect as soon as day 1 p.i. (Fig. 5G). Again, a good correlation existed between *in vivo* BLI and CFU at the endpoint, once more validating the BLI results (Fig. 5H). In conclusion, *in vivo* BLI provides a direct readout of the fungal load present over time instead of an indirect health effect and can therefore detect treatment efficacy for both triazole-susceptible and -resistant AF strains with improved sensitivity and at an early time point compared to health score and survival readouts.

## DISCUSSION

In this study, we developed the first bioluminescence readout to noninvasively visualize and quantify triazole-resistant and -susceptible *A. fumigatus* fungal burden kinetics over several days and in real time in *Galleria mellonella* larvae, with reliable differences being observed by as early as day 1 after infection, at which point survival and health score analyses failed to show differences. For establishing this BLI-supported model, we first optimized the substrate dose that can be administered daily without significant effects on larval health (40 $\mu$g/g D-luciferin). While higher D-luciferin doses provided stronger *in vivo* BLI signals, they were less well tolerated, most likely due to the observed D-luciferin accumulation in larvae. However, even at the low dose of D-luciferin, the resulting BLI signal significantly distinguished between various infecting doses of red-shifted luciferase-expressing *A. fumigatus* strains in *G. mellonella*

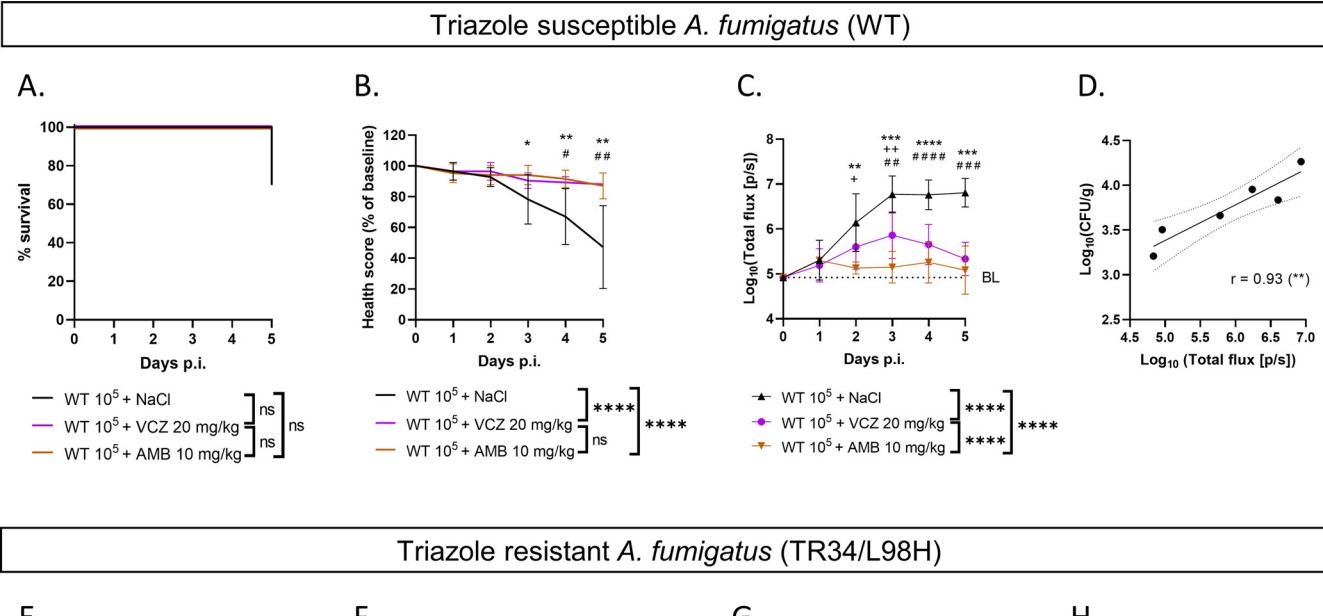

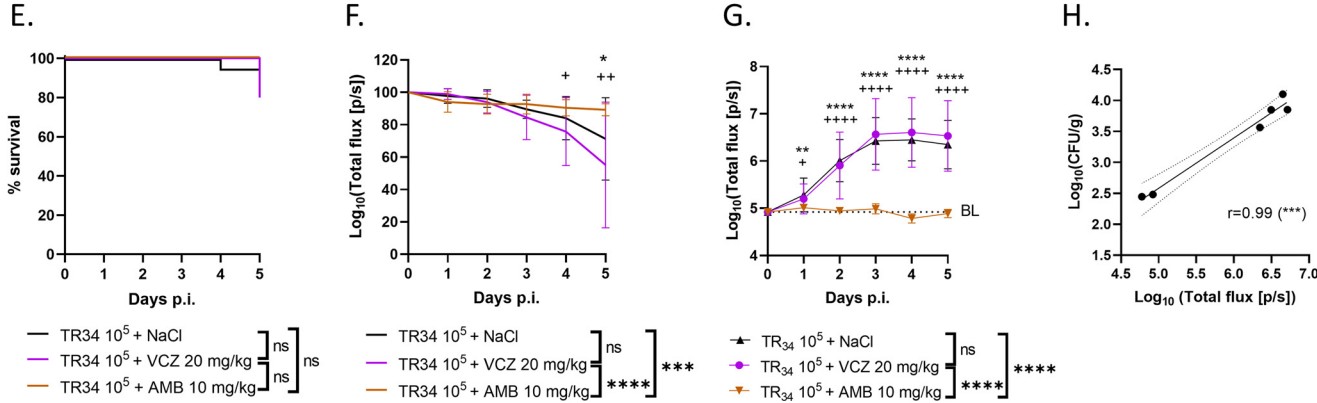

**FIG 5** Antifungal screening of triazole-susceptible and -resistant AF strains in the *G. mellonella* infection model. Shown are survival rates, health scores, BLI signals, and endpoint correlations between *in vivo* BLI signals and the respective CFU of larvae infected with $10^5$ conidia of wild-type (WT) (triazole-susceptible) *Aspergillus fumigatus* (A to D, respectively) and larvae infected with $10^5$ conidia of AF TR$_{34}$/L98H (triazole-resistant *A. fumigatus*) (E to H, respectively) over 5 days after infection. BL (baseline) represents the background signal (dotted lines in panels C and G). r, Pearson correlation coefficient. Data are means ($\pm$SD) ($n$ = 10 for all longitudinal data; $n$ = 2 per group for endpoint correlations). Statistics on the graph refer to differences on individual days between voriconazole (VCZ)- and amphotericin (AMB)-treated ("+"), between AMB- and NaCl-treated ("*"), and between VCZ- and NaCl-treated ("#") larvae. Statistics in the keys refer to longitudinal differences over 5 days. */#/+, $P < 0.05$; **/##/++, $P < 0.01$; ***/###/+++, $P < 0.001$; ****/####/++++, $P < 0.0001$; ns, nonsignificant.

at an earlier time point than the other available readouts, survival and health index scores ($\pm 2$ days). Furthermore, BLI presented a large dynamic range of infection detection when quantitatively compared to CFU counts, yet it was not as sensitive as CFU counts for low fungal burdens at the onset of infection. In addition, we validated and showed the potential of our novel BLI readout for antifungal screening against triazole-susceptible and -resistant *A. fumigatus* strains *in vivo*. Ultimately, by relying on the direct quantification of the amount of the fungus in each larva by BLI instead of indirect and subjective health effects that may be caused by infection and repeated injections, intra- and interexperimental larva variability were reduced, increasing the significance of our readouts.

Survival is by far the most commonly used readout in *G. mellonella* larvae for swiftly testing novel antifungal compounds but provides a binary output and is insensitive for all nonlethal differences between groups. In order to detect more subtle health differences, Loh and colleagues introduced the health index scoring system that, apart from survival, also takes into account activity, cocoon formation, and melanization (33). While this health scoring index allows the estimation of larval health over time, it still requires high fungal doses to induce visible health impairment in a short time after infection. Also, the levels of mobility and melanization are open to interpretation and can induce interobserver

variability, especially between different laboratories. We compared survival and health index scores to BLI over a time span of 5 days postinfection, as often used in the literature. While survival and health index scores need 3 to 5 days to show significant differences between groups, we demonstrated that BLI can already distinguish between groups shortly after the onset of infection. Moreover, BLI does not require larval death as the endpoint, which, depending on the dose of infection, may not be reached within a 5-day observation period. Over time, there is a flattening of the BLI curves of the highest-infected groups as larvae die when they reach a high fungal burden and therefore no longer contribute to the increase in the BLI signal. This introduces an increasing survival bias, which is unavoidable with this type of data. Overall, BLI can distinguish different fungal loads in *G. mellonella* with equal or better sensitivity and in a shorter time span than survival and health index score analyses. The advantage that BLI offers is of particular benefit when working with lower inoculum doses that do not affect overall larval health, as BLI is the only readout that is still able to detect differences.

As confirmed by our Bland-Altman comparisons and Pearson correlation analyses, an important advantage of BLI is its direct reflection of the fungal load in comparison to CFU analyses. Although survival and health scoring readouts also provided reasonably good measures for infection studies, they are only indirect consequences of the fungal burden, with a delayed readout. Thus, BLI is more convenient and reliable for the determination of the effect size of antifungals toward mouse studies. Moreover, a direct readout of the fungal burden broadens the applications of the *G. mellonella* model toward single-species monitoring of the bioluminescent fungus in coinfections with other fungal species or strains (e.g., mixed triazole-susceptible and -resistant *A. fumigatus* infections, which is currently being investigated by our group) or other microorganisms. In that way, potential synergistic or antagonistic behaviors of the bioluminescent fungus in combination with other microorganisms can be observed over time, along with how this interaction affects treatment efficacy. This would be impossible using only survival, health score, or even endpoint readouts such as CFU. The same advantage applies to unwanted microorganisms that might be present in non-research-grade *G. mellonella* larvae; by using a bioluminescent experimental strain, it is certain that the signal measured originates from the experimental infection. Consequently, if a larva dies without a major BLI signal, it can be excluded with reason, and it will not induce false variability. Our method also opens the door to multispectral bioluminescence imaging of coinfections in *G. mellonella*.

CFU determination is still considered the gold standard for absolute fungal quantification. However, in contrast to the longitudinal character of health index scoring and BLI, the CFU count is an endpoint measure that is unable to capture the dynamic aspect of infection in time and space. BLI combines this dynamic longitudinal aspect with fungal quantification because of its noninvasive character, and it is much less labor-intensive and faster than the evaluation of CFU. When comparing fungal quantifications by BLI and CFU over a large dynamic range ($10^3$ to $10^8$ spores per larva) on the day of infection, there is a good correlation between both readouts. This is taking into consideration the lower detection limit of BLI of around $10^4$ photons per s. As such, for very small amounts of fungi present and especially at the early stage of conidia that are just waking from dormancy, CFU determination has a higher sensitivity than BLI. However, when comparing both readouts in a complete 5-day experiment with a small dynamic range ($10^5$ spores per larva), the correlation weakens. The smaller dynamic range of the fungal load present in the larvae reveals a low capacity of CFU to distinguish between small differences in the fungal loads compared to BLI. This wide dynamic detection range of BLI compared to CFU can also be seen in *ex vivo* BLI, performed on the same homogenates as the ones plated for CFU counts, confirming that not the samples but the methodology led to these results. These methodological differences will result mainly from the fact that a fungal mycelium, formed during prolonged growth, will naturally lead to an underdetermination of the CFU compared to early time points at which individual spores or germlings are observed.

The first reported use of BLI of fungal infections in *G. mellonella* larvae was from Delarze et al. (34). They designed a bioluminescent *Candida albicans* strain expressing *Gaussia*

*princeps* luciferase and longitudinally monitored infection in the larvae by *in vivo* BLI using a luminometer until 3 days postinfection, with and without fluconazole (4 mg/kg) treatment. They demonstrated a good correlation between the *in vivo* BLI signal and CFU in nontreated larvae although only at 24 h p.i. Corresponding to our findings, their correlation also shows a higher sensitivity of CFU for lower fungal counts. However, they used *Gaussia princeps* luciferase with coelenterazine as a substrate. Coelenterazine needs methanol as a solvent, which may cause severe side effects from repeated injections. In addition, *Gaussia* luciferase produces primarily blue light, with poor tissue penetration and high background emission (34, 35). This causes challenges in translation to mice with deep-seated infection, such as in the lungs. By using a red-shifted firefly luciferase with D-luciferin, our model is more easily transferred to mouse models of AF infection, allowing the use of the same substrate throughout the preclinical pipeline (32). More recently, Milhomem Cruz-Leite et al. developed bioluminescent *Paracoccidioides brasiliensis* and *Paracoccidioides lutzii* strains expressing a red-shifted firefly luciferase and tested them in *G. mellonella* larvae using *in vivo* BLI although rather as a proof of concept (36). They performed BLI at 0 h, 24 h, and 6 days postinfection using a 10-times-lower concentration of D-luciferin than we did. While bioluminescence was observed, they did not quantify or analyze their BLI results or survival data over time, nor did they correlate their findings with CFU, so a comparison of their data with our results is difficult.

Altogether, we believe that the implementation of *in vivo* BLI in *G. mellonella* research can improve the quality and reproducibility of results by complementing or potentially replacing longitudinal health scoring and CFU readouts, on the condition that a bioluminescent strain is available. Our *in vivo* BLI method is versatile and can be adapted to different laboratory settings. Although we used self-bred larvae for our experiments, we show that equal results can be obtained in commercially bought non-research-grade larvae. Moreover, *in vivo* photon flux can also be measured by a luminometer instead of an IVIS Spectrum system (34). Another possibility would be to use fluorescence imaging (FLI) instead of BLI. FLI eliminates the need for substrate administration, but on the other hand, its signal-to-background capabilities for sensitive fungal burden detection and its applicability for treatment screening remain to be evaluated when red or near-infrared fluorescence-expressing azole-resistant and -susceptible strains would become available.

In conclusion, we successfully optimized and established the use of bioluminescence imaging to quantify the fungal burdens in individual *Galleria mellonella* larvae in real time and over the course of the experiment, with a focus on *A. fumigatus*, and showed its application toward antifungal efficacy screening in triazole-resistant and -susceptible infections. By providing a longitudinal readout that is quantitative and time-efficient, we combine the advantages of health index scoring and CFU determination in a single readout. In addition, BLI has a larger dynamic range than endpoint CFU counts and allows fungal loads to be distinguished earlier than with survival and health index scoring readouts, especially when working with low fungal doses. Our model provides the unique opportunity to longitudinally detect fungal burden kinetics in a direct way, thereby reducing the large variation known in the *G. mellonella* field and promoting interstudy comparisons and reproducibility. We believe that the use of BLI in *G. mellonella* for antifungal screening will contribute to the more successful translation of novel antifungal strategies from *in vitro* screenings to mice. We therefore consider this model especially relevant for the study of triazole-susceptible and, importantly, triazole-resistant *A. fumigatus* strains in order to target current therapeutic needs in the field.

## MATERIALS AND METHODS

**Bioluminescent *Aspergillus fumigatus* strains and culture.** We used previously validated triazole-susceptible (WT) (Af_luc$_{OPT\_red}$_WT) and triazole-resistant (TR$_{34}$/L98H) (Af_luc$_{OPT\_red}$_TR$_{34}$) *A. fumigatus* strains expressing a codon-optimized red-shifted firefly luciferase (32). The MICs of voriconazole (VCZ) for the bioluminescent WT and TR$_{34}$/L98H strains are 0.5 and 8 mg/L, respectively. Spores were cultured on Sabouraud agar plates containing chloramphenicol and incubated for 48 h at 37°C before they were harvested by adding 5 mL AD–0.1% Tween 80 (Sigma-Aldrich, St. Louis, MO, USA) and gently scraping the conidia off the surface with a sterile cotton swab. The resulting conidial suspension was filtered (11-$\mu$m-pore-diameter nylon membrane;

**TABLE 1** Health index scoring system for *G. mellonella* larvae[a]

| Category | Description | Score |
|---|---|---|
| Movement | No movement | 0 |
| | Minimal movement upon stimulation | 1 |
| | Movement when stimulated | 2 |
| | Movement without stimulation | 3 |
| Melanization | Completely black | 0 |
| | Black spots on brown larva | 1 |
| | ≥3 spots on beige larva | 2 |
| | <3 spots on beige larva | 3 |
| | No melanization | 4 |
| Survival | Dead | 0 |
| | Alive | 2 |
| Total score | | /9 |

[a]Adapted from reference 33. Total scores were converted to percentages.

Merck Millipore, Burlington, MA, USA) to remove spore clumps and hyphae, favoring the presence of single spores. The fungal suspensions were centrifuged, pelleted, and washed in sterile PBS to remove the Tween 80.

For experiments, *A. fumigatus* spore suspensions were counted using a Neubauer hematocytometer and diluted to the required conidial concentration in PBS. The spore count in the final suspension was confirmed by *in vitro* bioluminescence imaging (BLI) and CFU plating as described below.

***In vitro* and *ex vivo* bioluminescence imaging.** To confirm the relative spore count in the inoculum and larval homogenates, 10-fold serial dilutions were made in a black 96-well plate (Cliniplate; Thermo Scientific, Denmark), and 10% D-luciferin potassium salt (1.25 mg/mL in PBS; Promega, USA) was added. The BLI signal was read using an IVIS Spectrum imaging system (PerkinElmer, USA) by acquiring 5 consecutive images with an exposure time of 30 s (open filter, F/stop1, subject height of 0.5 cm, and medium binning). Living Image software (version 4.5.4; PerkinElmer, USA) was used to define regions of interest (ROIs) covering each well and to calculate the total photon flux (photons per second) per well; peak total fluxes were used for analysis and comparison. Experimental data were pooled only if the confirmed inoculum sizes were identical.

**CFU.** To determine the absolute viable spore counts in the inocula (CFU per milliliter) and larval homogenates (CFU per gram), 10-fold serial dilutions were made in a black 96-well plate (Cliniplate; Thermo Scientific, Denmark). Fifty microliters of each dilution was plated onto Sabouraud agar plates containing chloramphenicol and incubated at 37°C, and spores were counted after 48 h. Experimental data were pooled only if the anticipated inoculum doses were identical.

***Galleria mellonella* infection model.** Since we experienced a large amount of variability with externally purchased larvae, we set up our own controlled *Galleria mellonella* breeding. However, we showed that equal results in terms of survival, health score, and *in vivo* BLI readouts can be obtained using healthy commercially purchased non-research-grade larvae (see Fig. S4 in the supplemental material). Healthy 6th-instar larvae weighing 300 ± 50 mg with normal movement and no melanization were selected for the experiments. The larvae were randomly assigned to experimental groups ($n = 10$ per group) and housed individually in 12-well plates to provide sufficient space to move normally. They were kept at 37°C in the dark without food. Fungal inocula of 10 $\mu$L were administered via the last right proleg into the hemocoel using a Hamilton syringe (10 $\mu$L, model 701SN, 31 gauge; Hamilton Company, Switzerland). The larval health score (movement, melanization, and survival [33]) was assessed daily for 5 days after infection (Table 1). Negative controls were sham infected with PBS. Untouched larvae were kept in parallel as a quality reference for the batch of larvae used (not shown). At sacrifice/death, infected larvae were weighed and homogenized individually or pooled per group (Tissue Master homogenizer; Omni International, Tulsa, OK, USA) in 600 $\mu$L PBS per larva for fungal quantification by CFU plating and *ex vivo* BLI.

**Antifungal treatment.** For VCZ (Vfend; Pfizer, USA), a stock solution of 10 mg/mL was prepared in 0.9% sterile saline and adjusted to obtain a final dose of 20 mg/kg in 0.9% sterile saline when injecting 10 $\mu$L and assuming an average weight of 300 mg per larva. For amphotericin B (AMB) (Fungizone; Bristol Myers Squibb, Canada), a stock solution of 5 mg/mL was prepared in sterile AD and adjusted to obtain a final dose of 10 mg/kg in 5% glucose when injecting 10 $\mu$L and assuming an average weight of 300 mg per larva. All treatments were freshly prepared and administered daily by intrahemocoel injection alternately in the last left and right prolegs to prevent potential injury by repeated injection in the same proleg. Infected control groups were injected with the treatment vehicle, and noninfected control groups received the treatment as controls and to assess toxicity.

***In vivo* bioluminescence imaging.** BLI was performed daily from the baseline (before infection) until day 5 postinfection using an Ivis Spectrum imaging system (PerkinElmer). Larvae were injected with 10 $\mu$L D-luciferin (40, 400, or 4,000 $\mu$g/g in PBS for an average larva of 300 mg) into the hemocoel by alternating the last right or left proleg, transferred to a black 12-well plate with a transparent bottom (IBL Baustoff + Labor GmbH, Austria) at 37°C for 10 min, and imaged for bioluminescence light emission with the following settings: open filter, F/stop1, subject height of 0.5 cm, medium binning, and a 30s exposure time per image. Using Living Image software (version 4.5.4), the total photon flux (photons per second) per larva

was defined through circular ROIs of 2.5 cm in diameter covering each well. We also verified that any variability in peak photon fluxes among individual experiments that used the same anticipated inoculum size was actually due to variations in the infectious doses as determined by CFU and *in vitro* BLI.

**Statistical analysis.** All statistical analyses were performed using GraphPad Prism version 8.0.2 (GraphPad Software, USA). The log rank (Mantel-Cox) test was used for survival analysis. Longitudinal health scores and $\log_{10}$-transformed *in vivo* BLI data were analyzed by repeated-measures or mixed-effects two-way analysis of variance (ANOVA) with Tukey's correction for multiple comparisons to detect significant differences within an experiment and between groups at defined time points. Pairwise repeated-measures or mixed-effects two-way ANOVAs were performed to compare the slopes (interaction time and group) over time. Dead larvae were excluded from BLI and health score statistical analyses but were retained in the graph with their last values before death to avoid visual survival bias. The correlation coefficient, *r*, was computed using Pearson's or Spearman's correlation analysis depending on the parametricity of the data. A *P* value of $<0.05$ was considered significant.

**Data availability.** Data are available upon request.

## SUPPLEMENTAL MATERIAL

Supplemental material is available online only.

**SUPPLEMENTAL FILE 1**, DOCX file, 0.6 MB.

## ACKNOWLEDGMENTS

We thank Rob Lavigne and his team for introducing us to working with *G. mellonella* larvae, Annouschka Laenen and Margaux Delporte for their valuable biostatistical advice, Charles Van der Henst for the initiative of starting up our own *G. mellonella* breeding, and Julien Brillard, Gaetan Clabots and François Uytterhoeven for their useful advice and practical help in setting up the breeding. Bioluminescence imaging was performed at the Molecular Small Animal Imaging Centre (KU Leuven).

Conceptualization, E.V. and G.V.V.; data curation, E.V., L.M., and A.R.-S.; formal analysis, E.V. and L.M.; funding acquisition, G.V.V.; methodology, E.V. and G.V.V.; supervision, G.V.V., A.R.-S., and K.L.; writing—original draft, E.V.; writing—review and editing, E.V., L.M., A.R.-S., M.B., K.L., and G.V.V. All authors have read and approved the final version of the manuscript.

This work was supported by the Flemish Research Foundation (FWO) (grants 1506114N and G057721N) and KU Leuven internal funds (C24/17/061). E.V. received an FWO aspirant mandate (1SF2222N).

K.L. received consultancy fees from MRM Health and MSD, speaker fees from Pfizer and Gilead, and a service fee from Thermo Fisher Scientific and TECOmedical. The remaining authors report no conflict of interest.

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
