## [Reviewer comments · Microbiology Spectrum]

Microbiology Spectrum

Powerful and real-time quantification of antifungal efficacy against triazole-resistant and -susceptible *Aspergillus fumigatus* infections in *Galleria mellonella* by longitudinal bioluminescence imaging

Eliane Vanhoffelen, Lauren Michiels, Matthias Brock, Katrien Lagrou, Agustin Reséndiz Sharpe, and Greetje Vande Velde

Corresponding Author(s): Greetje Vande Velde, Katholieke Universiteit Leuven

Review Timeline:

Submission Date:	February 23, 2023
Editorial Decision:	April 10, 2023
Revision Received:	June 5, 2023
Accepted:	June 27, 2023

Editor: Patricia Albuquerque

Reviewer(s): Disclosure of reviewer identity is with reference to reviewer comments included in decision letter(s). The following individuals involved in review of your submission have agreed to reveal their identity: Nicolas Beziere (Reviewer #3)

Transaction Report:

DOI: <https://doi.org/10.1128/spectrum.00825-23>

April 10, 2023

Prof. Greetje Vande Velde
Katholieke Universiteit Leuven
Imaging and Pathology
Herestraat 49
O&N1 box 505
Leuven 3000
Belgium

Re: Spectrum00825-23 (Powerful and real-time quantification of antifungal efficacy against triazole-resistant and -susceptible *Aspergillus fumigatus* infections in *Galleria mellonella* by longitudinal bioluminescence imaging)

Dear Prof. Greetje Vande Velde:

Link Not Available

Sincerely,

Patricia Albuquerque

Journals Department
Reviewer comments:

Reviewer #1 (Comments for the Author):

The manuscript of Eliane Vanhoffelen and co-authors describes the development and validation of a luminescent based method to assess *A. Fumigatus* infections in *G. Melonella* larvae and its use for screening of antimicrobials. The reasons to develop such a method are compelling and the experiments are clearly designed and reported. As reviewer, I would suggest minor changes: 1) The paper lacks description of quantitative results in some sections. For example, page 6 line 127 authors says "large variability" :one might ask how large and how much they expected to reduce it. In page 11, "While CFU has a lower detection

limit, we showed that BLI has a larger dynamic range." : the description of the LOD and DR should be indicated.

2) The authors report that they use data above limit of detection but how much higher than detection limit they consider limit of quantification? In many bioluminescent methods using a luminometer a signal 3 times higher than LOD is considered reliable. What about their method?

3) Could the authors comment on the adaptation of their method in different well plate format (e.g. 24 well) or with the use of black plates with transparent bottom? Or the adaptation of the methods to standard luminometers?

This would give the reader the idea on robustness of the method.

4) In page 9 line 197 I think authors meant 400 µg/g, and 4000 µg/g, and not that 40 and 400 doses were not further evaluated.

Reviewer #2 (Comments for the Author):

Please see attached file containing review.

Reviewer #3 (Comments for the Author):

The manuscript titled "Powerful and real-time quantification of antifungal efficacy against triazole-resistant and -susceptible *Aspergillus fumigatus* infections in *Galleria mellonella* by longitudinal bioluminescence imaging" tackles the issue of the quality and accessibility of the animals models of fungal, or more specifically IPA, infections. This is commendable, as IPA was recently identified by the WHO as a major threat, and the general ethical restrictions on the use of animal model are increasing.

The manuscript is generally well written and easy to follow. There are however two major concerns that need to be addressed, alongside a few minor comments.

First and foremost, the modality chosen is BLI. While optical imaging is the obvious choice in such a model, it is very unclear to me why BLI specifically was chosen (benefits, drawbacks, etc) over standard fluorescence. (Red shifted) fluorescent strains of *Aspergillus fumigatus* have been published (see <https://doi.org/10.1016/j.ijmm.2014.08.009> for example), and the use of fluorescence over BLI would spare the larvae the injections, and the authors the need for the initial study presented in Fig. 1. Is it because of concerns with remaining signal obtained from the remaining tissue even after antifungal treatment? Bleaching? Limited availability of resistance strain with fluorescent expression? This should be presented and addressed in the introduction/discussion of the manuscript.

Second, and this would be pertinent to the discussion around the sensitivity of the approach, I think Fig. 5C and Fig. 5F should be analyzed more in depth. In particular, it seems the differences between the VCZ resistant strain and the WT strain appears only on day 3, so only once the therapy has stopped. Such a discussion is not apparent from the current text. This is particularly problematic as the initial aim was to provide a model to evaluate antifungals, yet their effect appear post treatment? A more in depth discussion on this topic should be provided.

Minor comments:

1. line 84-94: mention of classical animals models of IPA and their difficult handling, including the importance of neutropenia, could be added for context.
2. line 100: ref 16 only mentions *Aspergillus*. Please either amend the text to reflect it or add/change the reference.
3. line 158: not state of the art, but rather first line in the clinics (other antifungals are appearing, this could be misleading).
4. Fig. 2A: mixing WT and VCZ resistant strain in the plot is misleading. Please identify (color code?) the strains or separate the plots.
5. line 252: please double check all mentions of abbreviations. Here, VCZ and AMB are not defined (only in Materials and Methods).
6. all figures: it is unclear what the statistic "stars" refer to in the figure itself, below the graphs. Significance at the final timepoint?
7. line 312: Here, it is unclear if it is a mention of multispectral luminescence, or single species monitoring in multiple infections. Please clarify.
8. line 354: "and" should not be in italic.

Staff Comments:

Preparing Revision Guidelines

To submit your modified manuscript, log onto the eJP submission site at <https://spectrum.msubmit.net/cgi-bin/main.plex>. Go to

Author Tasks and click the appropriate manuscript title to begin the revision process. The information that you entered when you first submitted the paper will be displayed. Please update the information as necessary. Here are a few examples of required updates that authors must address:

Please return the manuscript within 60 days; if you cannot complete the modification within this time period, please contact me. If you do not wish to modify the manuscript and prefer to submit it to another journal, please notify me of your decision immediately so that the manuscript may be formally withdrawn from consideration by Microbiology Spectrum.

The manuscript Spectrum00825-23 entitled “Powerful and real-time quantification of antifungal efficacy against triazole-resistant and -susceptible *Aspergillus fumigatus* infections in *Galleria mellonella* by longitudinal bioluminescence imaging” presents an interesting assay for rapid determination of antifungal efficacy *in vivo* against triazole-resistant and -susceptible strains of *A. fumigatus*. The manuscript is well written, I have only one major comment regarding the *G. mellonella* model and four minor comments for the figures/statistical analyses.

As the authors pointed out in the introduction, one obvious disadvantage of the *G. mellonella* model is the uncontrolled quality/condition of larvae provided by commercial suppliers. Therefore, (line 144-145) “additional dynamic readouts that bypass health parameters are thus necessary to unlock the full potential of this promising model”, which is BLI in this study. On the other hand, (line 415-416) “since we experienced a lot of variability and reproducibility issues with externally purchased larvae, we set up our own controlled *Galleria mellonella* breeding”. As a potential user of the *G. mellonella* model (who might not be able to set up in-house breeding), it seems to me that even though the authors emphasized the reliability and sensitivity of their BLI assay, it might be difficult to get reproducible data if commercially available larvae are used for experiments. It would be interesting to see the comparison between purchased larvae and in-house bred larvae concerning the reliability and sensitivity of the BLI assay vs. CFU assay (both bypass health parameters).

Minor comments:

Line 611: Regarding the Bland-Altman comparison in Fig 2A & 2D, could you please explain how you made an average / determined the difference of data with two different units (p/s and CFU/g)?

Line 214-215: “...in moderate- to high-dose infected larvae (10^5 - 10^8 conidia), a good correlation was observed between *in vivo* BLI and CFU counts for TR34/L98H (Fig 2B) and WT (Fig 2C)”, (line 611) the highest values of $\text{Log}_{10}(\text{CFU/g})$ in Fig 2B and 2C are around 4.5 and 5.0, respectively, how is the correlation between CFU and infectious dose? Are you sure it is CFU/g not CFU/mg?

Line 642-644: '+' indicated difference between 10^3 and 10^4 conidia per larva, '**' indicated difference between 10^4 and 10^5 conidia per larva, what does '#' indicate (it did not appear in Fig 4)?

Line 654-655: "No significant differences existed between NaCl and VCZ treated groups", then why there are significant differences indicated between these two groups in the legends of Fig 5B (***) and 5C (*)?

The manuscript titled "Powerful and real-time quantification of antifungal efficacy against triazole-resistant and -susceptible *Aspergillus fumigatus* infections in *Galleria mellonella* by longitudinal bioluminescence imaging" tackles the issue of the quality and accessibility of the animals models of fungal, or more specifically IPA, infections. This is commendable, as IPA was recently identified by the WHO as a major threat, and the general ethical restrictions on the use of animal model are increasing.

The manuscript is generally well written and easy to follow. There are however two major concerns that need to be addressed, alongside a few minor comments.

First and foremost, the modality chosen is BLI. While optical imaging is the obvious choice in such a model, it is very unclear to me why BLI specifically was chosen (benefits, drawbacks, etc) over standard fluorescence. (Red shifted) fluorescent strains of *Aspergillus fumigatus* have been published (see <https://doi.org/10.1016/j.ijmm.2014.08.009> for example), and the use of fluorescence over BLI would spare the larvae the injections, and the authors the need for the initial study presented in Fig. 1. Is it because of concerns with remaining signal obtained from the remaining tissue even after antifungal treatment? Bleaching? Limited availability of resistance strain with fluorescent expression? This should be presented and addressed in the introduction/discussion of the manuscript.

Second, and this would be pertinent to the discussion around the sensitivity of the approach, I think Fig. 5C and Fig. 5F should be analyzed more in depth. In particular, it seems the differences between the VCZ resistant strain and the WT strain appears only on day 3, so only once the therapy has stopped. Such a discussion is not apparent from the current text. This is particularly problematic as the initial aim was to provide a model to evaluate antifungals, yet their effect appear post treatment? A more in depth discussion on this topic should be provided.

Minor comments:

1. line 84-94: mention of classical animals models of IPA and their difficult handling, including the importance of neutropenia, could be added for context.
2. line 100: ref 16 only mentions *Aspergillus*. Please either amend the text to reflect it or add/change the reference.
3. line 158: not state of the art, but rather first line in the clinics (other antifungals are appearing, this could be misleading).
4. Fig. 2A: mixing WT and VCZ resistant strain in the plot is misleading. Please identify (color code?) the strains or separate the plots.
5. line 252: please double check all mentions of abbreviations. Here, VCZ and AMB are not defined (only in Materials and Methods).

6. all figures: it is unclear what the statistic “stars” refer to in the figure itself, below the graphs. Significance at the final timepoint?

7. line 312: Here, it is unclear if it is a mention of multispectral luminescence, or single species monitoring in multiple infections. Please clarify.

8. line 354: “and” should not be in italic.

30th May 2023

RE: Revised Manuscript Spectrum00825-23

Dear Dr Albuquerque,

We are pleased that the reviewers and editorial board appreciated the relevance of our findings in our manuscript titled “Powerful and real-time quantification of antifungal efficacy against triazole-resistant and -susceptible *Aspergillus fumigatus* infections in *Galleria mellonella* by longitudinal bioluminescence imaging” and grateful for the opportunity to submit a revised version. We have addressed the reviewers’ comments in a revised manuscript with changes highlighted in yellow, and summarized in the point-by-point responses below. We hereby want to thank the reviewers for their constructive feedback that helped us to improve the paper even further. We have high hopes that we could implement the reviewers’ feedback to their satisfaction and that you would consider our revised manuscript for publication in Microbiology Spectrum.

For all authors, sincerely yours,

Greetje Vande Velde, PhD.
Professor Faculty of Medicine, Dept. of Imaging and Pathology.
Molecular Small Animal Imaging Center (MoSAIC).
Email: Greetje.vandavelde@kuleuven.be

Point-by-point responses to reviewers’ comments

Reviewer #1:

We feel encouraged by the reviewer recognizing that the reasons to develop our methodology are “compelling” and for highlighting that “the experiments are clearly designed and reported”.

Minor comments:

- “The paper lacks description of quantitative results in some sections. For example, page 6 line 127 authors says "large variability": one might ask how large and how much they expected to reduce it. In page 11, "While CFU has a lower detection limit, we showed that BLI has a larger dynamic range." : the description of the LOD and DR should be indicated.”

Response: The “large variability” on page 6 line 128 in the introduction refers to a general problem among *Galleria mellonella* users. As explained in the following sentences of the manuscript, it comes on the one hand from the lack of research-grade larvae, introducing variability between batches of larvae, and on the other hand from the indirect and scale-based nature of the most commonly used readouts such as survival and health index. We have now clarified and specified the nature of the variability and how we expect to reduce it, namely by introducing research-grade larvae on the one hand, and on the other hand, by introducing a direct, unbiased readout for fungal burden.

We have similarly revised our variability statements in various sections to be more precise and quantitative where possible.

On page 10-11 (line 213-214 and 235-237) we added a quantification of the lower detection limit and dynamic ranges of BLI and CFU as per the request of the reviewer.

- “The authors report that they use data above limit of detection but how much higher than detection limit they consider limit of quantification? In many bioluminescent methods using a luminometer a signal 3 times higher than LOD is considered reliable. What about their method?”

Response: We quantified the baseline BLI signal in every experiment and all signal above baseline that could be visually verified, was considered reliable for fungal quantification. This visual verification was done pragmatically, and was possible because, differently from using a luminometer, by using a camera we have access to this additional information beyond the raw photon counts. We did therefore not employ a predefined n-fold increase in signal as a cut-off. Because we interpret our BLI signals on a log-scale and evaluate differences after log-transforming them, every significant increase above baseline is substantial. As illustrated in figure 2A (new) and B, a clear cut-off can be defined between signal below and above baseline. In figure 2C and D we show that all data that we defined to be above baseline in figure 2A and B, correlates well with CFU, proving that this pragmatic approach is reliable.

- “Could the authors comment on the adaptation of their method in different well plate format (e.g. 24 well) or with the use of black plates with transparent bottom? Or the adaptation of the methods to standard luminometers? This would give the reader the idea on robustness of the method.”

Response: We already used black plates with transparent bottoms, I added this in the materials (line 461). For housing of the larvae during 5 days post infection, we would recommend to keep them in 12- (or less) well plates, providing sufficient space to move normally (added in line 436-437). Imaging them in a 24-well plate instead of a 12-well plate is possible, and reading out bioluminescence data in a luminometer is certainly a valid alternative, supported by Delarze et al (line 353). A paragraph on the versatility of the model was added in the discussion (line 370-381).

- “In page 9 line 197 I think authors meant 400 µg/g, and 4000 µg/g, and not that 40 and 400 doses were not further evaluated.”

Response: We did mean to write between 40 and 400 µg/g; it was an explanation of why we did not look in more detail for the highest tolerated dose between 40 and 400 µg/g, to further increase the photon flux. However, since it is not of significant importance for the manuscript and might cause confusion, we left this sentence out.

Reviewer #2:

We appreciate that the reviewer recognizes our bioluminescent methodology as an “interesting assay for rapid determination of antifungal efficacy *in vivo* against triazole-resistant and -susceptible strains of *A. fumigatus*” and are pleased to read that he or she found the manuscript “well written”.

Major comment:

- “As the authors pointed out in the introduction, one obvious disadvantage of the *G. mellonella* model is the uncontrolled quality/condition of larvae provided by commercial suppliers. Therefore, (line 144-145) “additional dynamic readouts that bypass health parameters are thus necessary to unlock the full potential of this promising model”, which is BLI in this study. On the other hand, (line 415-416) “since we experienced a lot of variability and reproducibility issues with externally purchased larvae, we set up our own controlled *Galleria mellonella* breeding”. As a potential user of the *G. mellonella* model (who might not be able to set up in-house breeding), it seems to me that even though the authors emphasized the reliability and sensitivity of their BLI assay, it might be difficult to get reproducible data if commercially available larvae are used for experiments. It would be interesting to see the comparison between purchased larvae and in-house bred larvae concerning the reliability and sensitivity of the BLI assay vs. CFU assay (both bypass health parameters).”

Response: We have now added a comparison between equally infected self-bred larvae and a qualitative batch of store-bought larvae, showing that they give the same results in survival, health score and BLI (Figure S4, reference in line 432-434 of methods section). Therefore, potential users of our model can also use store-bought larvae from any breeder capable of delivering consistently qualitative batches of larvae.

We consider a batch of larvae qualitative when no melanization is visible and when no fungal co-infection becomes apparent upon plating the larval homogenates on Sabouraud agar for CFU.

Minor comments:

- “Line 611: Regarding the Bland-Altman comparison in Fig 2A & 2D, could you please explain how you made an average / determined the difference of data with two different units (p/s and CFU/g)?”

Response: Aiming at comparatively evaluating how BLI and CFU counting perform to quantify fungal burden over a dynamic range of time and inoculum size, we took the difference and average of the numerical values of BLI and CFU, and compared them making abstraction of their measurement units by expressing the resulting values in arbitrary units (A.U.), as added on the graphs (Figure 2B and F). Indeed, given that in theory, a Bland-Altman comparison is typically made for measures expressed in the same unit, with our biostatistician we have considered an alternative which involves a unit-less measure through normalizing the log-transformed BLI and CFU data using their Z-scores (number of standard deviations that a value is away from the average). Since this made the graphs unnecessary complicated to interpret, we decided to keep the graphs and the comparison in A.U. To even better graphically represent how BLI and CFU counts compare dynamically over a range of burden and time, we added two panels in Figure 2 (A and E) in which we show the BLI and CFU data of the Bland-Altman plots separately, in their original units, as complementary quantitative information to the Bland-Altman plots in arbitrary units. We believe these graphical representations explain the data in the best possible way to the reader.

- “Line 214-215: “...in moderate- to high-dose infected larvae (10^5 - 10^8 conidia), a good correlation was observed between *in vivo* BLI and CFU counts for TR34/L98H (Fig 2B) and WT (Fig 2C)”, (line 611) the highest values of $\text{Log}_{10}(\text{CFU/g})$ in Fig 2B and 2C are around 4.5 and 5.0, respectively, how is the correlation between CFU and infectious dose? Are you sure it is CFU/g not CFU/mg?”

Response: We are sure it is CFU/g, but indeed we systematically find lower CFU counts in the homogenates than would be expected based on the inoculum, counted in a Burkler chamber. Already in the inoculum, we find approximately 100 times lower counts on CFU than under the microscope. Since *in vitro* BLI signals of inoculum and homogenates do reflect the same total amount of conidia, we believe this is a systematic methodological shortcoming of CFU counting.

- “Line 642-644: ‘+’ indicated difference between 103 and 104 conidia per larva, ‘**’ indicated difference between 104 and 105 conidia per larva, what does ‘#’ indicate (it did not appear in Fig 4)?”

Response: Thank you for this remark, we removed the ‘#’ in the figure legend (Fig 4).

- “Line 654-655: “No significant differences existed between NaCl and VCZ treated groups”, then why there are significant differences indicated between these two groups in the legends of Fig 5B (***) and 5C (*)?”

Response: With this sentence we meant that no significant differences existed between these groups on individual days, as measured by repeated measures or mixed effects two-way ANOVA of all groups per individual timepoint (indicated by stars on the graphs itself). However longitudinally, when taking into account the course of infection over 5 days, significant differences could be detected (indicated by stars in the legend). For this we performed pairwise repeated measures or mixed effects two-way ANOVA including all timepoints. Since we repeated these experiments and replaced figure 5 by new data (cfr. major comment of reviewer 3), this comment is not applicable anymore, but we made sure to indicate the meaning of the statistics on the graphs versus in the legend more clearly.

Reviewer #3:

We appreciate that the reviewer recognizes the need to tackle “the issue of the quality and accessibility of the animal models of fungal, or more specifically IPA, infections” and that he or she finds the manuscript “generally well written and easy to follow”.

Major comments:

- “First and foremost, the modality chosen is BLI. While optical imaging is the obvious choice in such a model, it is very unclear to me why BLI specifically was chosen (benefits, drawbacks, etc) over standard fluorescence. (Red shifted) fluorescent strains of *Aspergillus fumigatus* have been published (see <https://doi.org/10.1016/j.ijmm.2014.08.009> for example), and the use of fluorescence over BLI would spare the larvae the injections, and the authors the need for the initial study presented in Fig. 1. Is it because of concerns with remaining signal obtained from the remaining tissue even after antifungal treatment? Bleaching? Limited availability of resistance strain with fluorescent expression? This should be presented and addressed in the introduction/discussion of the manuscript.”

Response: We think FLI could also be used instead of BLI, with the necessary adaptations, and we would be interested to run a side-by-side comparison in the near future to evaluate the questions raised by the reviewer, however we did not do this yet. The reasons why we started with BLI are multiple. Most importantly, BLI is based on an active enzymatic reaction in which the fungal-produced luciferase uses oxygen and ATP as co-factors to convert luciferin into oxyluciferin, thereby producing photons. The requirement of ATP and oxygen implies that the fungus has to be alive to generate a bioluminescent signal, in contrast to FLI which relies only on the presence of the fluorescent protein, and then indeed the half-life of the fluorescent protein comes into play. Therefore, dead fungal spores or hyphae could still produce a fluorescent signal upon excitation and this could, at least in theory, hamper real-time treatment screening.

We believe BLI has a better sensitivity than FLI because of lower background signal and therefore higher signal-to-background, which is especially important in the early timepoints of our application given the proven advantage that BLI can detect treatment effects as soon as at day 1 or 2 post infection, while we have not seen (published) evidence of such sensitivity with FLI. The increased sensitivity of BLI compared to FLI becomes even more relevant when taking into account the translatability of our *G. mellonella* model towards imaging-based mouse models of invasive aspergillosis, where photons coming from deep-seated lung lesions have to cross multiple tissue-layers before detection and the advantages of BLI are even higher. In order to be able to use the same methodology throughout translation of *in vitro* to *G. mellonella* to mice, BLI is for us the obvious first choice.

Finally, to the best of our knowledge no red or NIR fluorescent *Aspergillus fumigatus* strain exists that carries the most common mutations in the *cyp51A* gene to generate triazole-resistance, leaving BLI as the only option at this moment to study azole-resistant infections and treatment response.

We do agree that FLI could have advantages in certain setting as laid out by the reviewer and it may be well worth evaluating also FLI in a similar setting. We have added the most important points on the consideration of the optical imaging modalities that can be applied in the discussion (line 370-381).

- “Second, and this would be pertinent to the discussion around the sensitivity of the approach, I think Fig. 5C and Fig. 5F should be analyzed more in depth. In particular, it seems the differences between the VCZ resistant strain and the WT strain appears only on day 3, so only once the therapy has stopped. Such a discussion is not apparent from the current text. This is particularly problematic as the initial aim was to provide a model to evaluate antifungals, yet their effect appear post treatment? A more in depth discussion on this topic should be provided.”

Response: Indeed, the treatment effect of VCZ in the WT strain is rather small and the BLI curves only visually separate around day 2-3. The main reason for this is probably the low VCZ dose we used, 10 mg/kg, while the clinically used dose is almost double. Also, basing ourselves on a publication of Jemel et al (*J. Fungi* 2021, 7(12), 1012; <https://doi.org/10.3390/jof7121012>), we stopped treating after 48 h pi, causing the treatment effect to stay small. As our goal for this study was to evaluate the capability of BLI to reveal subtle treatment effects, we did not seek to establish a treatment protocol that would clear the infection early on. Nevertheless, we are confident that with a higher dose, our BLI-method would pick up a stronger treatment effect earlier on. Indeed, we have repeated the experiment with a higher dose of VCZ (20 mg/kg) and treated daily until experimental endpoint on day 5 pi, thereby strengthening the message of figure 5. We made use of this opportunity to also perform an endpoint validation of the *in vivo* BLI results of the treatment effect against CFU of larval homogenates and have included this extra data in the manuscript (new Fig.5).

Minor comments:

- “line 84-94: mention of classical animals models of IPA and their difficult handling, including the importance of neutropenia, could be added for context.”

Response: We emphasized the complexity of the model in the context of immunosuppression in line 92.

- “line 100: ref 16 only mentions Aspergillus. Please either amend the text to reflect it or add/change the reference.”

Response: We added other references.

- “line 158: not state of the art, but rather first line in the clinics (other antifungals are appearing, this could be misleading).”

Response: Adapted

- “Fig. 2A: mixing WT and VCZ resistant strain in the plot is misleading. Please identify (color code?) the strains or separate the plots.”

Response: We identified the WT and resistant strains by using dots or triangles on the graph respectively (adapted both panel A and B for consistency).

- “line 252: please double check all mentions of abbreviations. Here, VCZ and AMB are not defined (only in Materials and Methods).”

Response: Thank you for this remark, we added the abbreviations in line 259.

- “all figures: it is unclear what the statistic “stars” refer to in the figure itself, below the graphs. Significance at the final timepoint?”

Response: Thank you for pointing this out, we added the meaning of the stars in the graph versus in the legend in the manuscript (Fig 4 and 5); the statistics in the figure itself refer to differences at individual days whereas statistics in the figure legends refer to longitudinal differences over 5 days. It is interesting to add both since it not only tells you if you have differences over the 5-day course of infection, but also what the earliest timepoint is at which significant differences can be detected.

- “line 312: Here, it is unclear if it is a mention of multispectral luminescence, or single species monitoring in multiple infections. Please clarify.”

Response: In the first place this is meant towards single species monitoring in multiple infections, because the bioluminescent strains we use in this manuscript all have the same red-shifted bioluminescent spectra. However, this can of course be expanded towards multispectral imaging by using multispectral fungal strains and spectral unmixing. We clarified this in the manuscript (line 322 and 331-332)

- “line 354: “and” should not be in italic.”

Response: Adapted (now line 364)

June 27, 2023

Prof. Greetje Vande Velde
Katholieke Universiteit Leuven
Imaging and Pathology
Herestraat 49
O&N1 box 505
Leuven 3000
Belgium

Re: Spectrum00825-23R1 (Powerful and real-time quantification of antifungal efficacy against triazole-resistant and -susceptible *Aspergillus fumigatus* infections in *Galleria mellonella* by longitudinal bioluminescence imaging)

Dear Prof. Greetje Vande Velde:

Your manuscript has been accepted, and I am forwarding it to the ASM Journals Department for publication. You will be notified when your proofs are ready to be viewed.

Sincerely,

Patricia Albuquerque
Editor, Microbiology Spectrum
